# Host responses and viral traits interact to shape the impacts of climate warming on highly pathogenic avian influenza in migratory waterfowl

Claire S. Teitelbaum[1,2,3]*, Michael L. Casazza[4], Cory T. Overton[4], Elliott L. Matchett[4], Diann J. Prosser[5]

1 NASA Ames Research Center, Moffett Field, California, United States of America, 2 Bay Area Environmental Research Institute, Moffett Field, California, United States of America, 3 U.S. Geological Survey, Georgia Cooperative Fish and Wildlife Research Unit, Warnell School of Forestry and Natural Resources, University of Georgia, Athens, Georgia, United States of America, 4 U.S. Geological Survey, Western Ecological Research Center, Dixon, California, United States of America, 5 U.S. Geological Survey, Eastern Ecological Science Center, Laurel, Maryland, United States of America

* claire.teitelbaum@gmail.com

## Abstract

Emerging infectious diseases pose threats to wildlife populations, as exemplified by recent outbreaks of avian influenza viruses in wild birds. Climate change can affect infection dynamics in wildlife through direct effects on pathogens (e.g., environmental decay rates) and changes to host ecology, including shifting migration patterns. Here, we adapt an existing mechanistic model that couples migration and infection to study how traits of highly pathogenic avian influenza (HPAI) viruses contribute to HPAI outcomes in migratory waterfowl, then apply this model to explore potential impacts of climate change on HPAI dynamics. We find that the simulated impacts of HPAI on the host population under baseline climate conditions varied from no impact to 100% mortality, depending on viral traits. In most cases, traits related to transmission (i.e., contact rates, shedding rates) were more important for HPAI establishment probability, infection prevalence, and mortality than were other viral traits (e.g., environmental temperature sensitivity, cross-protective immunity). We then simulated the effects of climate change (i.e., altered temperature regimes) on HPAI dynamics both via viral environmental decay and via changes in bird migration phenology. In these simulations, we found that a 9-day advancement in spring migration timing increased the duration of HPAI outbreaks by increasing time birds spent at their breeding grounds, leading to higher mortality and fewer infections. In contrast, increased viral decay in warmer years had a smaller, but opposite impact. These patterns depended on the primary transmission mode of HPAI (i.e., direct vs. environmental) and its sensitivity to environmental temperatures. Together, these results suggest that climate change is likely to increase the impacts of HPAI on waterfowl populations if HPAI relies

**Data availability statement:** All code written in support of this publication and data necessary to reproduce results are available at Zenodo [94] https://doi.org/10.5281/zenodo.13355279.

**Funding:** The U.S. Geological Survey Environmental Health Program supported this work (DJP). The funders had no role in study design, data collection and analysis, decision to publish, or preparation of the manuscript.

**Competing interests:** The authors have declared that no competing interests exist.

strongly on direct transmission and birds advance their spring migration. Further integrating host-viral co-evolution and other climatic changes (e.g., salinity, humidity) could provide more precise predictions of how HPAI dynamics could change in the future.

## Author summary

Infectious diseases can cause major problems for wildlife. Recently, avian influenza viruses have caused significant mortality in some wild bird populations. These impacts might be increasing because of changes in the virus itself, and/or because of changes in the environments where wild birds live. Here, we use a set of simulations to show that these viruses have more severe impacts on wild bird populations when they are more easily transmitted, infection lasts longer, and prior infection provides little immunity. We also show that the effects of avian influenza on wild bird populations could shift with climate change, but whether climate warming reduces or intensifies these effects depends on how the virus and wild birds respond to warmer temperatures. Further experimental and observational research on virus traits and wild bird responses could improve our understanding of how influenza might respond to climate change.

## Introduction

Novel diseases have emerged at increasing rates in the last century [1], posing urgent threats to wildlife, livestock, and human health [2,3]. Among the many causes of this acceleration is climate change, which affects disease transmission directly and through changes to species' distributions [4–6]. Agricultural intensification and increased livestock production, which influence contact rates of humans and wildlife with livestock, are also important contributors to disease emergence [7]. In wildlife, the impacts of highly pathogenic avian influenza (HPAI) have grown steadily over the last several decades [8] and the ongoing outbreak of HPAI H5 clade 2.3.4.4b, which emerged in Eurasia in 2020 and in the Americas in 2021, has caused substantial mortality in wild birds as well as in some mammals. HPAI is defined by its ability to cause disease in poultry; until recently mortality events were concentrated in poultry, with occasional outbreaks in other wild bird species [9]. Recent mortality events have been particularly severe in colonially nesting seabirds, but some waterfowl, which were previously largely unaffected, have also experienced substantial mortality [10,11]. Many of these novel attributes could be linked to the evolution of higher virulence and slower environmental decay rates in currently circulating HPAI viruses [12–14]. Evidence from mechanistic models indicates that transmission mode, cross-protective immunity, and stochasticity are all important for the dynamics of avian influenza viruses (AIVs) in poultry and wild birds [15–19]. Therefore, understanding the roles of HPAI viral traits and host responses for HPAI dynamics in

wild bird populations can help us forecast how these dynamics might continue to change in the future, as these viruses continue to evolve and adapt to new hosts, new environments, and new transmission modes.

Climate change could impact HPAI dynamics through multiple mechanisms [20]. AIVs are transmitted at least partially through the environment when hosts ingest contaminated water or soil. Viruses usually decay faster at warmer temperatures, extreme pH, and higher salinity [21]. For temperature, there is also a trade-off between decay rates at cool temperatures and temperature sensitivity, such that viruses that decay more slowly at lower temperatures tend to be more temperature-sensitive [22,23]. Climate change might therefore reduce viral infectivity in the environment by increasing water temperatures and salinity [20]. Climate change could also impact HPAI dynamics via changes to avian migration patterns and habitat use. Over the last two decades, some bird populations have advanced their spring migration timing as vegetation greens up earlier on their northern breeding grounds [24]; this change in phenology could affect density-dependent transmission processes and increase time spent at high-latitude breeding grounds, where temperatures are cooler and thus environmental virus loads might be higher [20].

In addition to these impacts of climate change on host migration and viral decay rates, models show that habitat loss can promote influenza transmission in migratory hosts by increasing group sizes at remnant sites [25,26]. Wetland degradation and phenological mismatch between wetland suitability (e.g., rainfall) and bird migration might also lead to increased reliance on agricultural and human-dominated habitats by waterfowl, promoting contact between waterfowl and poultry [27–29]. Finally, changing migration phenology can make it more difficult to synchronize conservation efforts with migration [27,30], thus altering avian spatial ecology and demography. However, despite these many hypothesized impacts of global change on wild bird populations, we have little empirical evidence and few quantitative predictions for how climate change will impact avian influenza dynamics in wild birds (but refer to [25,31]).

Here, we use a mechanistic model to explore (1) the characteristics of HPAI viruses that most strongly influence their invasion (i.e., establishment), persistence, and impacts in migratory waterfowl, and (2) the potential effects of climate change on the dynamics and outcomes of HPAI. We measure both short-term outcomes (i.e., invasion probability, peak infection prevalence) and long-term outcomes (i.e., persistence probability, mortality, mean infection prevalence). Our model represents a population of migratory Pacific greater white-fronted geese (*Anser albifrons sponsa*; Fig 1A, B). Geese are an important taxon in HPAI ecology because they are extreme long-distance migrants that can disperse AIVs across large spatial scales [32] but are also more likely to show clinical signs of infection compared to many ducks [33] and can shed HPAI for longer [34]. They therefore represent a taxonomic group that is both potentially a vector and a victim of HPAI outbreaks [35], and a group where there is ample evidence of temperature-related changes to migration phenology over the last few decades [36,37]. This subspecies exhibits high fidelity to breeding sites (Yukon Delta) and winter sites (Sacramento Valley) [38]. Although these geese are a particularly amenable system, the model could also generalize to other latitudinally migratory waterfowl. Using our model, we explore how a severe climate change scenario impacts the dynamics of HPAI viruses with different traits, specifically temperature sensitivity and reliance on direct vs. environmental transmission. We also evaluate how HPAI dynamics change if climate change impacts viral decay in the environment, migration phenology, or both. We expected that climate change would reduce HPAI effects for environmentally transmitted strains when climate change increases viral decay rates and would increase HPAI effects for directly transmitted strains when climate change affects migration phenology.

## Methods

### Model framework

**Epidemiology.** Our model is based on compartmental epidemiological models of AIV developed by Brown *et al.* [16,39]. This model has previously been used to explore the roles of cross-immunity and cross-species transmission in driving HPAI and low pathogenic avian influenza (LPAI) dynamics in a migratory shorebird-waterfowl system [16,39]. The model includes seasonal host migration, multiple AIV strains (strain 1: LPAI and strain 2: HPAI), and multiple transmission

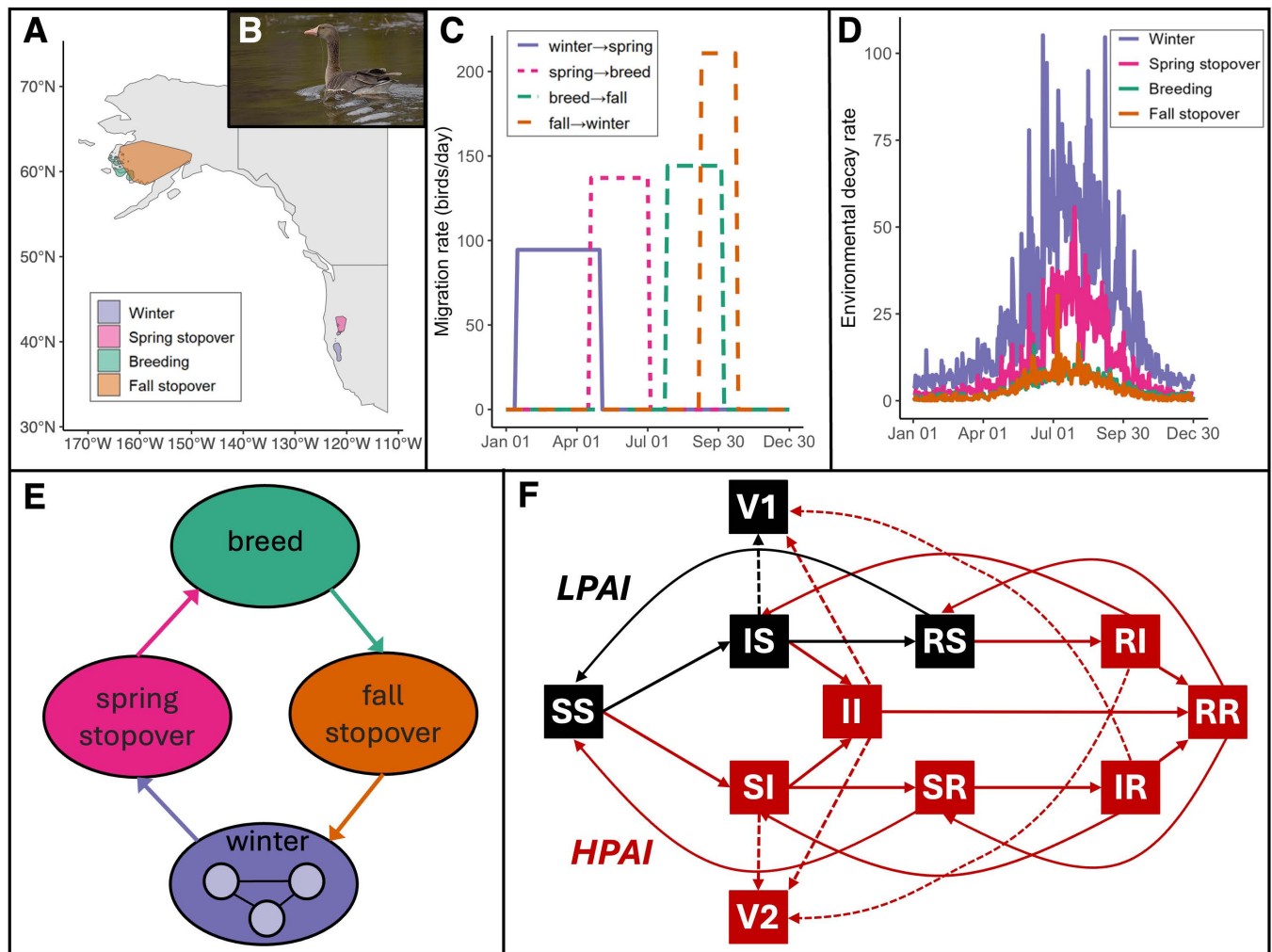

**Fig 1. Model structure and data inputs. (A)** Locations of seasonal sites derived from GPS tracking data. Basemap from Natural Earth in public domain (https://www.naturalearthdata.com/about/terms-of-use/). **(B)** Greater white-fronted goose. Photo by M. Lewandowski in public domain (https://npgallery. nps.gov/AssetDetail/f5338022-357c-4d27-bbf6-f2dd75af6542). **(C)** Timing of seasonal migration, as used in simulations, derived from goose tracking data. **(D)** Viral decay rates in the environment at each site. Daily air temperatures at each seasonal site, derived from the ERA5-Land reanalysis [46], inform viral decay rates in the model. **(E)** Conceptual diagram of sites and movement. **(F)** Box-and-arrow diagram of transmission. In each box, the first letter indicates LPAI infection status (Susceptible, Infectious, or Recovered) and the second indicates HPAI infection status. Viral reservoirs V1 and V2 represent LPAI and HPAI environmental reservoirs, respectively. Dashed lines represent viral shedding into the environment. Environmental exposure is not shown, but V1 and V2 contribute to transmission of the virus (e.g., V1 contributes to transition of SS to IS). Mortality is not shown but occurs in all classes, and at an additional rate from classes SI, II, and RI. Black boxes and arrows show compartments relevant in a single-strain LPAI model. The full diagram (including both red and black boxes and arrows) shows classes and transitions for a two-strain model. AIV transmission and dynamics occur simultaneously at all sites.

modes (Fig 1F). We modify the original model for a single species by adding infection-induced mortality (to reflect the ability of HPAI to cause mortality), a fourth season/site (to reflect goose migration patterns), multiple wintering sites (to reflect within-winter movements [40]), and an external environmental input of the virus (to represent shedding by infected heterospecifics or conspecifics). Compartments in the model are defined by their infection status (**S**usceptible, **I**nfectious, or **R**ecovered) with respect to each AIV strain and by their location (one of six sites; Fig 1A, E). Hosts can become infected via contact with an infectious host (at rate $\beta$) or via contact with virions in the environment (governed

by the environmental uptake rate $\rho$; Fig 1F). An environmental reservoir (**V**) exists at each site; the reservoir grows as infectious hosts shed virus (at rate $\omega$) and shrinks at a temperature-dependent environmental decay rate ($\eta$; Fig 1D and S1 Table). Infectious hosts recover from infection (at rate $\gamma$) and recovered hosts eventually lose immunity (at rate $\varepsilon$). We assume homogeneous mixing within each site but no contact among hosts at different sites. Co-infections are possible but are limited by cross-immunity from infection (i.e., individuals that are infectious or recovered from one strain are less likely to become infected with the other; $\psi_1$). We assume that HPAI can cause mortality, which is mediated by a second cross-protection parameter (i.e., reduced mortality rate in birds that are infectious or recovered from LPAI infection; $\psi_2$). We assume that parameters representing viral and host traits are fixed for a single parameterization of the model (i.e., no evolution or behavior change occurs). Full systems of equations are provided in Supplementary Methods (S1 Text); parameter ranges and their sources are provided in S1 Table.

**Seasonal migration.** We model migration across four seasons: winter, spring stopover, breeding, and fall stopover. Migration is unidirectional (e.g., wintering to spring stopover, not spring stopover to wintering) and each migration occurs across a range of dates (i.e., not all individuals move at once, Fig 1C). Spring stopover, breeding, and fall stopover each consist of one site, while winter consists of three sites; birds move among the three sites throughout the winter season, representing within-winter movements of this population [38,40]. Migration timing is parameterized using GPS telemetry data for Pacific greater white-fronted geese [41]. Briefly, we identified seasonal sites from GPS data by segmenting goose tracks algorithmically and manually into six classes (breeding, winter, fall and spring migration, fall and spring stopover), following Teitelbaum *et al.* [42]. We used these segmented tracks to identify the range of dates that geese move between sites (Fig 1C). To define the spatial location of each site, we used kernel density estimates to identify breeding and wintering areas and minimum convex polygons to identify fall and spring stopover areas (Fig 1A). These analyses were implemented in R using the *amt* and *adehabitatHR* packages [43–45]. For more detail, refer to Supplementary Methods (S1 Text).

We extracted daily mean temperatures at each site from the European Reanalysis-Land (ERA5-Land) product, which provides temperature estimates at ~9 km spatial resolution [46]. These temperature data were used to parameterize time- and site-specific viral decay rates in the environment (Fig 1D).

## Parameterization and model implementation

We implemented stochastic realizations of the model using the *adaptivetau* package in R [47]. This package uses the Gillespie algorithm with adaptive tau leaping [48], which efficiently simulates a stochastic process by adapting the time step of simulation depending on the current transition rates between classes and the probability that a state variable will reach a critical value (i.e., 0). In other words, the algorithm uses larger time steps (and thus simulates more quickly) when rates of change are slower and extinction is less likely. In this stochastic framework, rate parameters (e.g., mortality rate) are used to determine the probability of an event occurring in each time step; rate parameters themselves remain fixed or vary deterministically over the course of a simulation. We chose to use a stochastic version of the model because prior models have shown that stochasticity is important for AIV dynamics [49] and because we were interested in viral extinction, which is difficult to model using a deterministic framework.

Before introducing HPAI, we first modeled circulation of LPAI in the goose population. This initial step is designed to reflect realistic conditions in waterfowl populations, where LPAI is endemic, and to define baseline conditions for HPAI introduction, including LPAI infection prevalence and immunity, population size, and population distribution (i.e., host abundance at each site) (S1 Fig). We tested a range of LPAI virus parameters based on prior models and experimental data from literature (S1 Table, [50]). We used an initial population size of 5,000 individuals, which represents ~1% of the total population size of Pacific greater white-fronted geese [51] and therefore a group size that might share the same wintering, stopover, and breeding sites. For some parameters (i.e., recovery rate, waning immunity rate, lifespan), parameter ranges are relatively well defined by empirical studies (e.g., 3-fold difference in estimates of waning immunity

rate), whereas others (i.e., contact transmission rate, infectious dose) are poorly defined in natural systems and therefore require larger ranges in sensitivity analyses (S1 Table). We included a relationship between environmental decay at 0°C and temperature sensitivity, based on experimental data from Handel *et al.* [22]. Host and viral parameters were equal at all sites, except the contact transmission rate ($\beta$), which was half as large at the breeding site, to account for reduced movement and reduced inter-family contact during nesting [16], and the reproduction parameter (*b*), which was nonzero only at the breeding site. We ran five replicate six-year simulations per parameter set to ensure that infection and population size outcomes were stable (e.g., that birth rates and natural mortality produced no net change in population size). We found that most viral extinctions occurred within two years, so six years was sufficient to measure stable-state dynamics (if they existed). We then calculated summaries of LPAI dynamics (i.e., mean infection prevalence, peak infection prevalence, timing of infections) to identify a set of traits (hereafter a "strain") that produced realistic outcomes [33,52–57]. Criteria for selecting the baseline strain were: peak infection prevalence <0.2; extinction probability over six years <50%; mean annual peak infection prevalence >5%; annual mean infection prevalence <5%; and infection prevalence peaks in winter [33,52–57]. After selecting the focal strain, we ran 50 replicate simulations to provide starting conditions for HPAI simulations (S2 Fig).

After establishing this baseline, we introduced one individual infected with HPAI into the population (S1 Fig). Similar to our approach for LPAI, we examined a range of parameter values for HPAI. Because HPAI traits are less well understood, we established some HPAI trait value ranges relative to the selected LPAI strain; for example, we considered direct transmission rates between 1/10 and 1000 times the LPAI direct transmission rate (S1 Table). Although we refer to these parameter sets as "strains," they represent host as well as viral traits (e.g., host immune function impacts infection duration). We considered eight HPAI traits: direct transmission rate ($\beta_2$), recovery rate ($\gamma_2$), shedding rate ($\omega_2$), HPAI-induced mortality rate ($\nu_2$), cross-immunity from infection ($\psi_1$), cross-protection from mortality ($\psi_2$), environmental decay rate ($\eta_2$), and shedding rate of heterospecifics ($\zeta_2$). We used a random Latin hypercube design to sample 10,000 parameter combinations from the 8-dimensional space. This method assumes that parameters are independent; although some parameters might be related in reality (e.g., shedding rate and contact transmission rate [58]), this approach allowed us to explore the effects of each HPAI trait individually. To explore the relative importance of direct and environmental transmission, we also modeled strains with only one transmission mode by including parameter sets where $\beta_2 = 0$ or $\omega_2 = 0$, $\zeta_2 = 0$. For each parameter set, the HPAI-infected individual was introduced at each of five time-location combinations (S2 Table), for a total of 50,000 parameter sets. We ran each simulation to an endpoint of six years and performed five replicate simulations for each parameter set.

Finally, to explore the effects of climate change on HPAI dynamics, we selected focal HPAI strains that represented different combinations of transmission mechanisms ($\beta_2$, $\omega_2$) and environmental temperature sensitivity ($\eta_2$) (S3 Table and S3 Fig). We fixed other parameters within ranges with the most empirical support ($24 < \gamma_2 < 73; 5 < \frac{\gamma_2}{\nu_2} < 40; 0.1 < \psi_1 < 0.9; 0.1 < \psi_2 < 0.9$; S1 Table) and arbitrarily selected an HPAI strain from the HPAI sensitivity analyses that met these criteria. We then defined strains using all pairwise combinations of five values each of $\beta_2$, $\omega_2$, and $\eta_2$ from across their ranges (i.e., 0%, 25%, 50%, 75%, and 100% quantiles; S3 Table). For all simulations, we introduced HPAI on the breeding grounds on May 27 (*t*=0.4).

We studied the effects of climate change on HPAI dynamics via (1) changes in viral dynamics in the environment, (2) changes in migration phenology, and (3) the combination of the two (S4 and S5 Fig). We assumed that migration advanced 1.8 days for every degree increase in temperature at the breeding grounds, relative to average May temperatures on the breeding grounds from 2016-2022 (the upper estimate for the observed rate of advancement in greater white-fronted geese [36]). We used future temperature scenarios from CMIP6 ensembles [59,60] at the same sites as described above under a severe climate change scenario: Shared Socioeconomic Pathway (SSP) 5-8.5. To simplify analysis, we used projections for 2095, which was the warmest projected year in the 2025–2100 period. We also ran a "baseline" scenario using historical temperature data from 2020 (as used in HPAI and LPAI simulations above). We ran each simulation

to an endpoint of six years and performed 20 replicate simulations for each parameter set. Annual temperature dynamics were fixed within each parameter set (i.e., a simulation for 2095 used 2095 climate data for all six years).

## Analysis

For each HPAI simulation, we calculated six metrics of interest. First, we measured whether a virus successfully invaded into the population, defined as causing >100 infections, a breakpoint in the data above which strains were likely to cause more infections and persist for longer (S6 Fig). Then, for strains that successfully invaded, we calculated four outcomes: peak infection prevalence (i.e., maximum over six years), mean infection prevalence over six years, mortality (proportional change in population size at the end of the simulation), and virus persistence (i.e., whether virus was present in hosts at the six-year endpoint of the simulation).

To examine the impacts of HPAI traits on infection outcomes under baseline climate conditions, we used generalized additive models (GAMs [61]). GAMs are conceptually similar to linear models, but can include non-linear effects of independent variables, and initial data exploration indicated that many of these relationships were non-linear. We ran a separate model for each outcome variable. First, we fit a binomial GAM for invasion probability (a binary variable). Then, for strains that successfully invaded, we fit a GAM for each of the four continuous outcomes (peak prevalence, mean prevalence, mortality, persistence probability). We used a beta distribution with a logit link for prevalence and mortality outcomes because these outcomes were bounded and could be defined on the interval (0, 1). For persistence, we used a binomial GAM. In all cases, the independent variables in each model were: smoothing splines for each of the eight HPAI traits (log- or inverse-transformed as appropriate, S1 Table), smoothed interactions between each pair of HPAI traits, and a parametric term for the location and date of introduction. All smoothing splines were fit using a thin plate regression spline and a maximum of five knots, and we used penalized regression splines (i.e., integrated model selection) to decrease model complexity [62]. GAMs were implemented using the *mgcv* package in R [63].

We evaluated models using the built-in *gam.check* function in *mgcv*, which provides standard model diagnostics (i.e., residual plots). We also evaluated model fit by calculating relative root mean squared error (rRMSE, i.e., RMSE standardized by the mean of the response variable) for models of continuous variables and area under the receiver operating curve (AUC) for binomial models. We also simulated an additional 200 HPAI time series and evaluated model predictive ability by comparing rRMSE and AUC between test and training data sets. Finally, we calculated a metric of variable importance for each variable in each GAM, defined as the size of the range of predicted values across the range of a variable or pairwise combination of variables, with all other variables held at their median values. For example, to calculate variable importance for $\beta_2$, we evaluated the model at 100 values of $\beta_2$ between 0 and 36, with all other parameters held at their medians, then calculated the difference between the smallest and largest of these predicted values (on the link – i.e., logit – scale). Finally, we used min-max scaling to scale variable importance between 0 and 1 for each model to facilitate comparisons across models.

To examine the effects of climate change on HPAI dynamics, we measured the same five outcomes (i.e., invasion probability, peak infection prevalence, mean infection prevalence, mortality, and persistence) for each simulation. Because the parameters we used consistently produced outbreaks at the breeding site, we also calculated two additional metrics: peak infection prevalence at the breeding grounds and the duration of the outbreak at the breeding grounds. We defined the beginning and end of the outbreak as the date when >25 individuals were infected at the breeding site; informal sensitivity analysis showed that our results were not sensitive to this arbitrary threshold (tested values between 10 and 200). We used generalized linear models (GLMs) in which the response variable was the outcome of interest and the predictor variables were the strain (a categorical variable combining the values of $\beta_2$, $\omega_2$, and $\eta_2$) and its interaction with the climate impact (i.e., none/baseline, virus only, migration only, virus and migration). As for the HPAI simulations above, we used a binomial GLM for invasion probability, but we used a gaussian distribution for peak prevalence, mean prevalence, and mortality, because we found that using a beta distribution provided no better fit to the data. For continuous outcomes, we

only included parameter sets for which HPAI invaded in at least 50% of simulations. Most strains that successfully invaded persisted for all six years, so we did not model persistence probability.

## Results

### HPAI invasion, persistence, and impacts

We simulated 10,000 potential combinations of HPAI viral traits (referred to as "strains"; S1 Table), each of which was introduced at five date-location combinations. Of these, 61% successfully invaded the migratory goose population (i.e., caused at least 100 infections, S6 Fig). GAMs of all five HPAI outcomes had high predictive ability on both the training and test data, although models of invasion probability, persistence probability, and mortality generally performed better than those for peak and mean infection prevalence (S4 Table and S7 Fig).

In our models of invasion probability, we found that the direct transmission rate ($\beta_2$) was the most influential viral trait for determining invasion across the range of parameter values examined (Fig 2). With other parameters held at their median values, invasion probability ranged from 0 to 1 across the range of values of $\beta_2$. At the lowest values of $\beta_2$ ($\beta_2 \approx 10^{-2}$), the virus was only able to invade when shedding rates were very high ($\omega_2 \approx 10^9$; Fig 4A). No other viral traits were able to overcome direct transmission rates below $10^{-2}$. Accordingly, among strains with only direct transmission ($\omega_2 = 0$, $\zeta_2 = 0$), 75% invaded, whereas only 7% of those with only environmental transmission ($\beta_2 = 0$) invaded (S8 Fig). When direct transmission rates were low enough that invasion probabilities were <1 ($10^{-1.5} < \beta_2 < 10^{-0.5}$), recovery rate ($\gamma_2$; Fig 4B), HPAI-induced mortality rate ($\nu_2$), and cross-immunity ($\psi_1$) also influenced invasion probability, such that invasion was more likely when recovery was slower, mortality rates were lower, and cross-immunity was weaker (S9 Fig). At high shedding rates, environmental decay ($\eta_2$) influenced invasion probability, such that strains that decayed more slowly at low temperatures were more likely to invade (Fig 5). Higher cross-protection from mortality slightly increased HPAI invasion probability when HPAI-induced mortality rates were very high (e.g., increase in invasion probability from 0.85 ($\psi_2 = 0$, $\nu_2 = 120$, $\beta_2 = 0.38$) to 0.87 ($\psi_2 = 1$, $\nu_2 = 120$, $\beta_2 = 0.38$)), but had negligible effects otherwise. Similarly, higher shedding rates of heterospecifics ($\zeta_2$) increased invasion probabilities, but only when viral decay rates in the

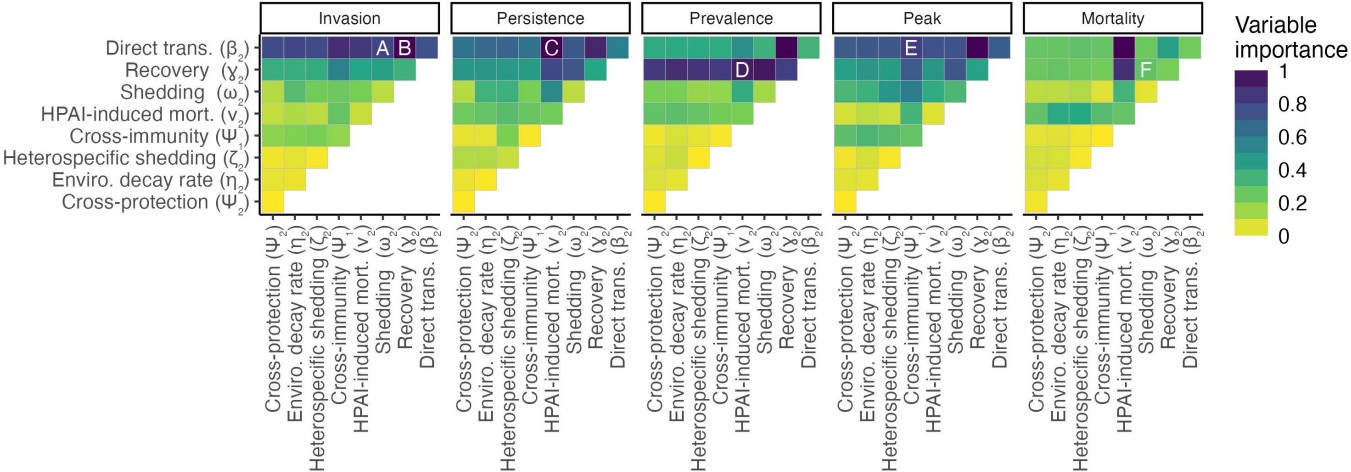

**Fig 2. Parameter importance for five outcomes of HPAI introduction.** Colors indicate variable importance, which is the expected magnitude of variation in an outcome across the range of a parameter or pair of parameters, with all other parameters held at their medians. Legend is binned for clarity, but colors are continuous. Cells along the main diagonal indicate the importance of each parameter alone. Letters in each inset plot correspond to panels in Fig 4. Variable importance is scaled between 0 and 1 to be comparable across outcomes. Note that variable importance is related to uncertainty in parameter values, such that parameters with larger (i.e., less well defined) ranges can produce higher variation in outcomes. However, some parameters with relatively well-defined ranges (e.g., recovery rate) still have high importance.

environment were very slow. The combination of introduction date and location also influenced invasion probability: HPAI invasion was least likely if it was introduced at the beginning of winter (S10 Fig).

Among strains that invaded, we examined four HPAI outcomes: persistence probability, mean infection prevalence over six years, peak infection prevalence, and proportional mortality. Across these simulations, 85% of strains persisted for the full six years, median mean infection prevalence was 0.05 (range: 0.00-0.10), median peak infection prevalence was 0.65 (range: 0.02-0.98), and median mortality was 0.17 (range: -0.21-0.97) (Fig 3).

Persistence probability was most sensitive to the HPAI-induced mortality rate ($\nu_2$), recovery rate ($\gamma_2$), and direct transmission rate ($\beta_2$), such that lower mortality rates, slower recovery, and higher direct transmission rates increased HPAI persistence probability (Fig 4C and S11). Recovery rate and HPAI-induced mortality rate were also important for determining mean infection prevalence (Fig 2 and S11), such that infection prevalence was highest when mortality and recovery rates were both slow (Fig 4D). Peak infection prevalence was generally most sensitive to the direct transmission rate (Fig 2), but high levels of cross-immunity ($\psi_1$) reduced peak prevalence, especially at intermediate direct transmission rates (Fig 4E). Finally, mortality increased with the HPAI-induced mortality rate (Fig 2), but also with higher shedding rates and slower recovery from infection (Fig 4F and S11).

Parameters related to environmental transmission and decay were important for some outcomes. For example, the viral decay rate ($\eta_2$) was more important for determining persistence than mean or peak infection prevalence. Strains

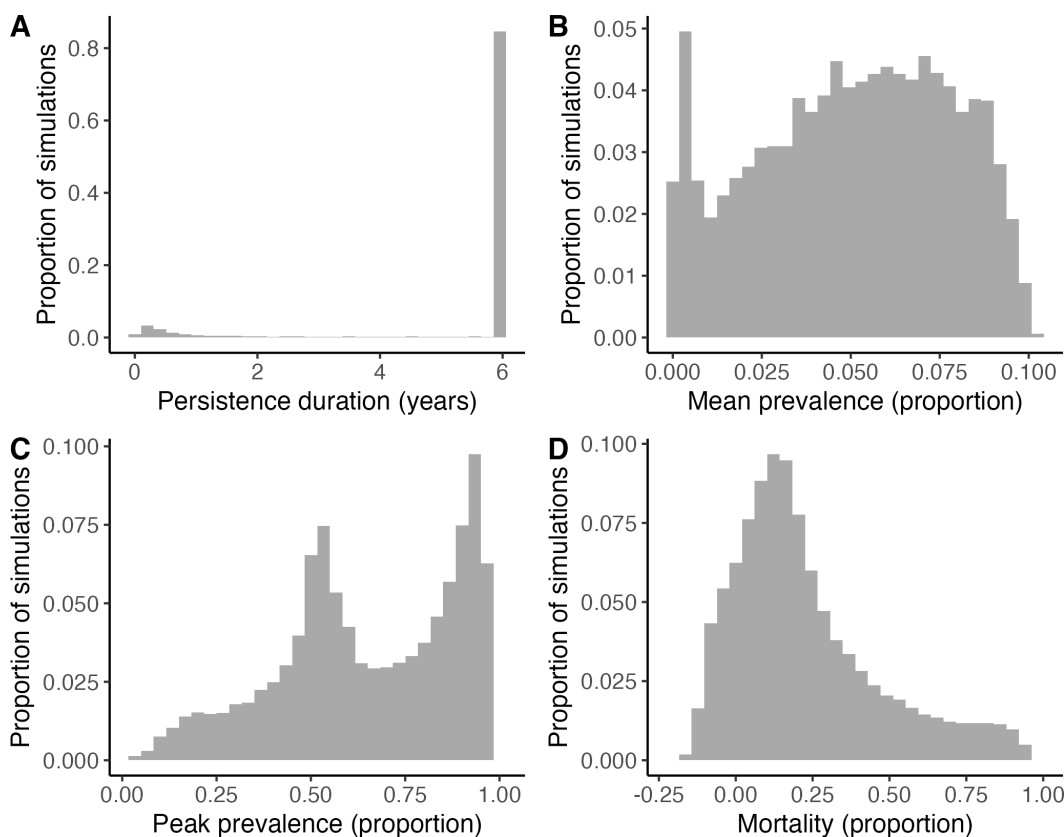

**Fig 3. Distributions of outcomes across simulations of HPAI introduction.** Each outcome was modeled separately as a function of HPAI traits. Only HPAI strains that successfully invaded (i.e., caused at least 100 infections) are shown. Because most strains either persisted for six years or became extinct within one year, we modeled persistence as a binary variable. Infection prevalence and mortality are measured as the proportion of the population. Negative values of mortality indicate population growth due to stochasticity.

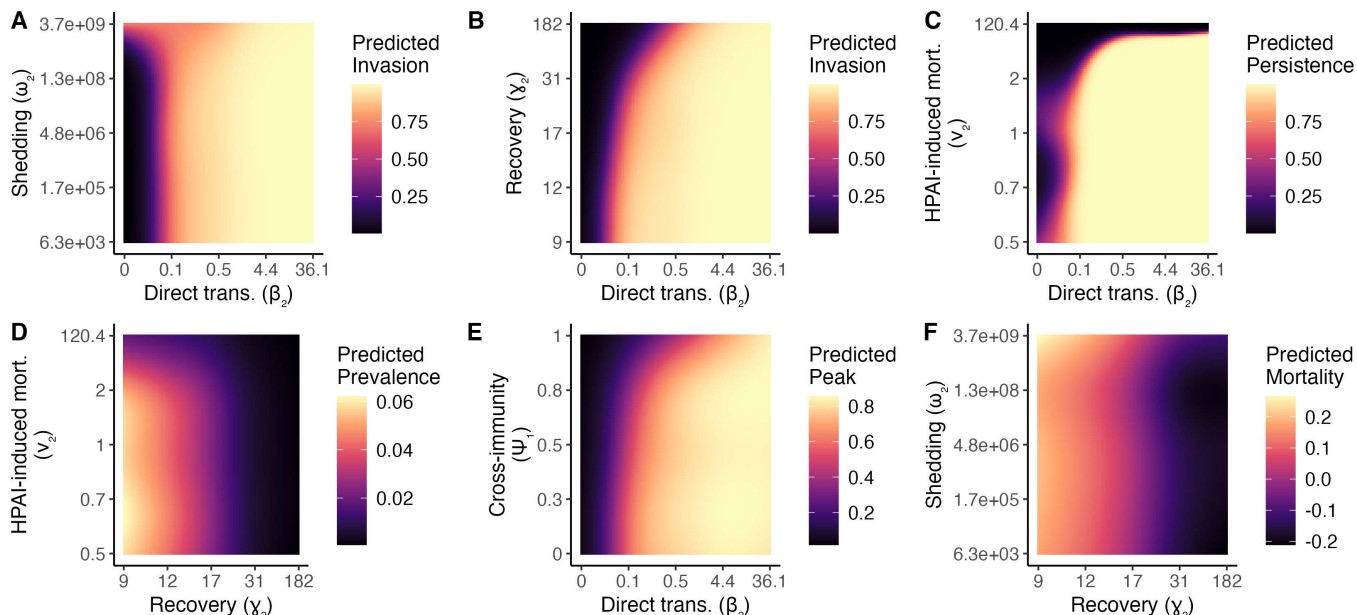

**Fig 4. Effects of focal parameter combinations on HPAI outcomes in the migratory goose population.** Each plot shows predicted outcomes from a generalized additive model that evaluated an outcome as a function of viral traits and time/location of introduction, including pairwise interactions. Outcomes modeled were (A, B) invasion probability (i.e., probability of causing >100 infections), (C) persistence probability, (D) mean infection prevalence over six years, (E) peak infection prevalence, and (F) proportional mortality. Negative mortality rates indicate population growth due to stochasticity. Note the transformations of some variables (inverse for $v_2$ and $\lambda_2$; $\log_{10}$ for $\beta_2$ and $\omega_2$). All continuous parameters not shown are held at their median values, except $\beta_2$, which is set at $10^{-1.5}$ to facilitate visualization of other parameters. Plots show fitted values for a strain introduced at the breeding site on September 13 (at the end of the breeding season).

that decayed more slowly at lower temperatures (and thus were more temperature-sensitive) produced more infections and caused more mortality (Fig 5). Environmental input of HPAI by heterospecifics ($\zeta_2$) was associated with increased mortality and infection prevalence, but only when viral decay rates were slow (S12 Fig). Among strains with only one transmission mode, invasion and persistence probabilities were higher for strains with direct transmission only, whereas strains with only environmental transmission had higher average mean infection prevalence, peak infection prevalence, and mortality (S8 Fig).

## HPAI dynamics under increasing temperatures

Climate projections predicted an increase in mean annual temperature of 6.52°C under SSP 5-8.5 (mean temperature in 2095 relative to 2020). Temperature increases were larger at the higher-latitude breeding and fall stopover sites (7.64°C and 7.60°C, respectively) than at wintering and spring stopover sites (4.02°C and 6.82°C, respectively). Average temperatures in May at the breeding site in 2095 were forecasted to be 5.1°C warmer than in 2020; in our simulations, this corresponded with a 9-day advance in spring migration dates (S5 Fig).

When we simulated HPAI dynamics under climate change, we found differing effects of climate warming on HPAI dynamics depending on whether temperatures affected viral dynamics in the environment or bird migration phenology (Fig 6). The effects of climate warming on HPAI outcomes also differed among HPAI strains with different traits (i.e., strength of each transmission mode, temperature sensitivity). Invasion probability and peak infection prevalence were reduced under climate change, but only when temperature impacted viral survival in the environment and when direct transmission rates were low (S13 and S14 Figs). For example, the probability of invasion for a focal strain with only environmental

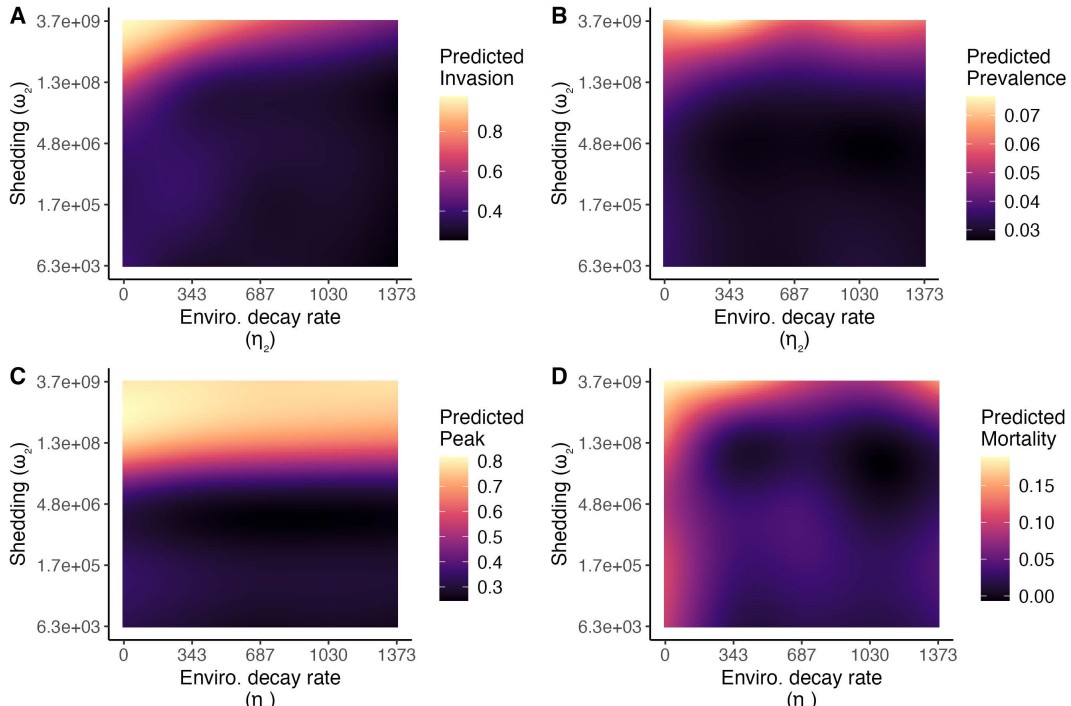

**Fig 5. Sensitivity of (A) invasion probability, (B) mean infection prevalence, (C) peak infection prevalence, and (D) mortality to environmental decay rates at 0°C ($\eta_2$) and the HPAI shedding rate ($\omega_2$).** Strains that survive longer at 0°C are more temperature sensitive. All outcomes except peak infection prevalence are more severe when shedding rates are high and environmental decay rates are low; peak infection prevalence is generally unaffected by the environmental decay rate. Each plot shows predicted outcomes from a generalized additive model that evaluated an outcome as a function of viral traits and time/location of introduction, including pairwise interactions. All continuous parameters not shown are held at their median values, except $\beta_2$, which is held at $10^{-1.5}$. Plots show fitted values for a strain introduced at the breeding site on September 13 (at the end of the breeding season). Note the $\log_{10}$ scale of the y-axes.

transmission and moderate environmental temperature sensitivity was 0.85 under baseline conditions and when climate change affected migration, but 0.62 when climate change affected viral survival in the environment. Mean infection prevalence tended to decline under climate change when temperatures impacted viral survival and to increase when temperature impacted migration phenology (Fig 6C), but expected changes were not biologically important (i.e., largest expected change in infection prevalence was 0.02%).

In place of effects on mean infection prevalence over the duration of the simulation (Fig 6C), we observed changes in mid-year dynamics of the virus. Specifically, when climate change impacted migration phenology, outbreaks at the breeding grounds occurred earlier but were slightly smaller at their peak (Fig 6A, 6B, and S3). These smaller outbreaks were in part due to increased mortality earlier in the outbreak, and accordingly total mortality was higher when climate change impacted migration phenology (Fig 6D). For example, for a strain with a high direct transmission rate and no environmental transmission, mortality increased by 4.5% compared to a baseline scenario, associated with a 9-day increase in outbreak duration and a slight decrease in outbreak size (0.2% decrease in peak infection prevalence). These impacts were more common in strains with higher direct transmission rates but were generally unaffected by a strain's environmental transmission rate or environmental temperature sensitivity (S15-S17 Figs). In contrast, when climate change impacted viral survival in the environment, outbreak duration and size were generally unaffected, but when they were, outbreaks were slightly larger in size and slightly shorter in duration (Figs 6, S15, and S16).

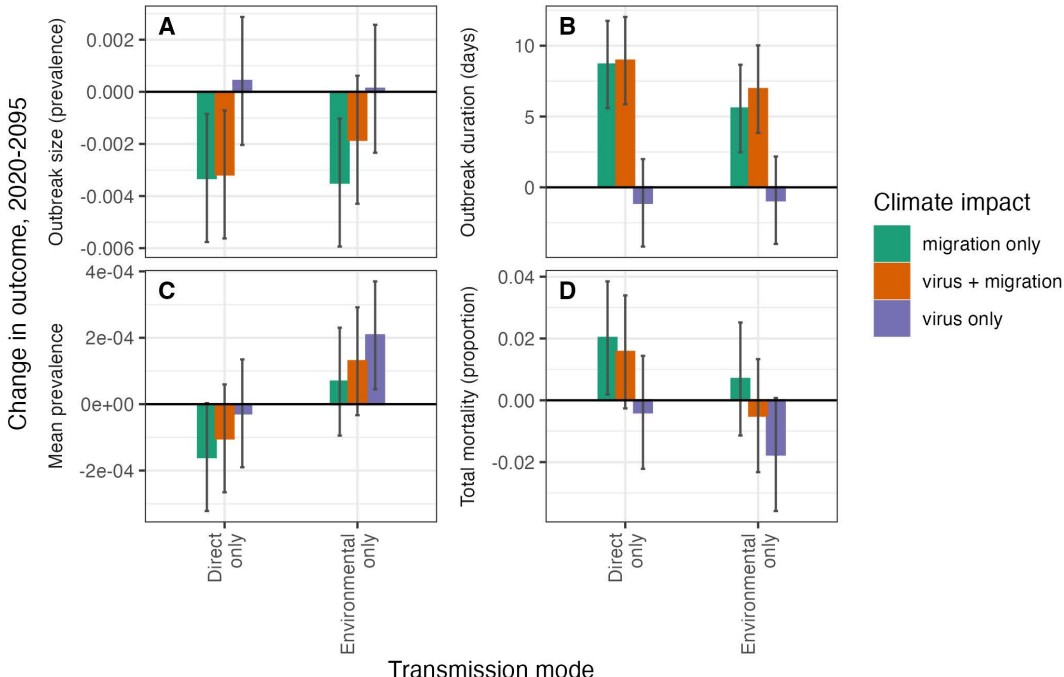

**Fig 6. Impacts of climate change depend on HPAI traits, climate effects (i.e., viral persistence, migration phenology, or both), and measured outcomes.** Bars show the expected change in each outcome variable between 2020 and 2095, based on climate projections from CMIP6 ensembles, estimated from a (generalized) linear model. Error bars show 95% confidence intervals. **(A)** Outbreak size is the peak infection prevalence on the breeding grounds in the second year after HPAI introduction. **(B)** Outbreak duration is the length of the breeding grounds outbreak in the second year after HPAI introduction. **(C)** Mean infection prevalence is the average proportion of the population infected over the six-year simulation. Note the very small effects (i.e., scale of the y-axis). **(D)** Total mortality compares the population size at the beginning and end of the six-year simulation. Two strains are shown here: both strains have a high environmental decay rate at 0°C and high temperature sensitivity (intercept $\eta_2 = 0.145$). The environmental-only strain has high environmental transmission rates ($\beta_2 = 0$, $\omega_2 = 3.69 \times 10^9$) and the direct-only strain has high direct transmission rates ($\beta_2 = 36.13$, $\omega_2 = 0$). For all parameter sets, refer to S15-S18 Figs.

## Discussion

Climate change can impact infectious disease dynamics through multiple mechanisms, including changes to host and pathogen ecology. Our results show that climate warming can have diverging effects on avian influenza dynamics, depending on whether temperature affects bird migration, viral decay rates in the environment, or both. Further, these impacts differ across influenza strains, depending on their transmission mechanisms and environmental temperature sensitivity. These results highlight the complexity of even simplified versions of natural systems, and thus the challenge of forecasting how the dynamics of wildlife diseases will change under future climate scenarios.

Under current (baseline) climate conditions, we found that HPAI viruses with a wide range of trait combinations could invade and persist in a migratory waterfowl population. For example, HPAI could invade and persist even when cross-immunity from prior LPAI infection was relatively high [16], as long as direct and/or environmental transmission rates were high enough [49]. After invasion, effects of HPAI on the goose population still depended on multiple viral traits, and notably on transmission parameters. For example, mean infection prevalence following invasion and total mortality were sensitive to the viral shedding rate (i.e., environmental transmission); these results support prior modeling studies showing that direct transmission is more important at the beginning of an outbreak, whereas environmental transmission contributes to endemicity and occasional outbreaks [49,64]. In the context of viral evolution and the ongoing HPAI panzootic, these results suggest that a shift from direct to environmental transmission could promote longer outbreaks, more mortality, and/

or HPAI endemicity in some populations [13,65]. Future studies that consider constraints on viral evolution (e.g., tradeoffs among traits) could further clarify how selection pressures, viral physiology, and environmental conditions could interact to promote or constrain environmental transmission rates of HPAI viruses.

In our models, a higher HPAI-induced mortality rate was associated with reduced mean infection prevalence and viral persistence probability, especially for directly transmitted strains (S8 Fig). This pattern, which corresponds with evolutionary theory [66,67], emerged because population sizes dropped below the critical community size for sustained transmission [64]. However, virus extinction came at a cost to host population sizes, including total population extinction in some cases (Fig 3). For multi-host pathogens like influenza viruses, these dynamics are important because species and age-classes differ in their infection tolerance [35,68,69], which can result in viral persistence in tolerant groups (i.e., reservoir species, adults) and repeated spillover to susceptible species or juveniles [32]. These dynamics could underly the repeated outbreaks of HPAI in threatened and highly impacted species like seabirds and raptors [10]. Species-level variation in infection tolerance is therefore an important component of infection outcomes, even for species with similar distributions, behavior, and exposure.

A combination of experimental and observational work informed parameter values in our models, but ranges were still relatively wide. Parameter ranges are often challenging to define in disease models, especially for parameters like the direct transmission rate ($\beta$), which encompasses multiple processes including social behavior, viral loads, and aerosol movement [70], and thus is usually "tuned" relative to other parameters rather than assigned a value based on empirical data. Other parameters, such as cross-reactive immunity, take a wide range of values in nature, depending on the pair-wise combination of viruses and their traits (e.g., structural similarity) [71]. Still others, including the recovery rate, waning immunity rate, and cross-immunity, are based on ample data and thus have defined ranges (e.g., duration of immunity 3 months-2 years) but still vary among and within individuals [72–74]. Avian influenza is among the best-studied wildlife diseases; a recent compilation of experimental infection data from wild avian influenza hosts moves the field towards a better understanding of means and variances in transmission parameters in controlled settings [50], but it also highlights strong taxonomic biases in studies (i.e., towards dabbling ducks) and a need to understand how these parameters change across natural contexts. A combination of experimental and field data could fill this gap. For example, modeling methods that are validated with cross-sectional and longitudinal sampling of wild hosts can identify the most likely transmission routes and parameter values in a given system [75]. Our results, which show that both direct and environmental transmission can produce persistent HPAI viruses, indicate that a better understanding of shedding rates, environmental decay rates, and the components of direct transmission could be priorities for further study.

In this study, HPAI temperature sensitivity in the environment was an important parameter for determining HPAI impacts under climate change. In general, HPAI strains that persisted for longer at low temperatures (and were therefore more temperature-sensitive) were more strongly impacted by climate change. Observational studies of where and how HPAI viruses persist in the environment [76,77], combined with experimental studies of the role of environmental exposure for HPAI transmission [13] and the temperature sensitivity of different viral strains [22], would help refine expectations for how climate change will affect HPAI dynamics. In addition to rising global temperatures, climate change is also impacting precipitation regimes, habitat quality, and other environmental parameters, whose effects on viral decay rates could differ from those of temperature. Experimental work that quantifies responses of AIVs to other climate-related characteristics, namely water pH and salinity [78], could also increase the accuracy of predictions of HPAI dynamics in the future.

This model presented a simplified version of a natural system to explore the roles of HPAI traits and climate change for viral dynamics in a migratory waterfowl population. Under baseline conditions, we effectively replicated LPAI infection dynamics in geese [56], even while excluding several mechanisms known to be important for AIV ecology, including demography (i.e., age structure, [74,79]), viral circulation among non-migrants on the wintering grounds [80,81], and non-homogeneous mixing among individuals at the same site. For example, geese forage in groups of familiar individuals [82], which could result in slower or smaller outbreaks relative to equal contacts between all individuals at a site

[83]. Future studies that build on this model [16,39] and others [25,64,72,73] could tease apart the relative importance of these mechanisms for HPAI dynamics under climate change, to further test the hypothesis that local dynamics are key for continental spread of AIVs [84]. These models would be strengthened by validation with empirical data on HPAI infection prevalence and its demographic impacts in goose populations, especially as surveillance efforts continue. Models that include demographic impacts of climate change (e.g., increased or decreased recruitment) will also be important, given that evidence for demographic impacts of climate change is growing [85–87]. In addition, the impacts of climate change on habitat availability for waterfowl can affect influenza dynamics [20], for example by concentrating animals at fewer sites [25]. Future studies that incorporate both spatial and temporal changes in migration would be valuable. As models become increasingly complex, and therefore less analytically tractable, individual-based models informed by animal telemetry data will be useful tools for understanding these mechanisms and their effects (e.g., [25]).

Our models were based on projected changes to temperature based on a single climate-change scenario and historical evidence of changes to migration. It is possible that these results would differ under different climate scenarios or with the inclusion of fine-scale data on microhabitats (e.g., temperature stratification in water, freeze-thaw cycles, fine-scale differences in temperature between sun and shade), which could provide thermal refugia for hosts or virions [88]. In addition, migration could adapt differently than observed, for example by advancing more than expected, changing in its variance, or altering the number or spatial configuration of sites; impacts might also differ in systems with alternative host migration biology, for example if hosts use multiple stopover sites [26], if HPAI infection affects migration rates [89], or if phenological mismatch due to climate change limits hosts' abilities to respond to infection [90]. Movement ecology of waterfowl is incredibly diverse, including non-migrants, nomads, and long-distance migrants; future empirical studies that use tracking data to study how individuals and populations respond to temperature and other climate variables would broaden our expectations for how climate change will affect disease dynamics across diverse species and populations. Further, HPAI viruses are continually evolving in natural systems [91]. Although our models considered a range of viral traits that represented potential outcomes of this evolution, future models that explicitly incorporate the processes of mutation and selection could provide more insight into the likely trajectory of these viruses, in particular how they might adapt to continuing shifts in host ecology and environmental conditions.

Infectious diseases have played important roles in host ecology and demographics over evolutionary history [92,93]. Climate change is impacting these dynamics by altering both host and pathogen ecology [4]. Changes to animal behavior in response to environmental change can affect disease outcomes, but these changes are difficult to predict because they depend on a suite of behavioral, ecological, and evolutionary processes that operate at different time scales. Our study, which integrates animal telemetry, climate models, and mechanistic models, shows that such modeling efforts can advance our understanding of how these processes fit together.

## Supporting information

**S1 Text. Supplementary methods.**
(DOCX)

**S1 Fig. Methods workflow.** See main methods for more details on data inputs, simulation methods, and analysis methods.
(PDF)

**S2 Fig. Baseline dynamics of low pathogenic avian influenza (LPAI) in the migratory goose population.** Each line represents a stochastic realization from the same parameter set. Parameters are shown in S1 Table.
(PNG)

**S3 Fig. Example HPAI dynamics under climate change.** Each panel shows an example single simulation (of 20 runs for each HPAI strain). Columns show strains with different transmission modes (see Fig 6).
(PNG)

**S4 Fig. Viral decay rates under a future climate scenario.** The y-axis shows the rate of viral decay in the environment. Columns show values of $\eta_2$ and rows show sites.
(PNG)

**S5 Fig. Migration phenology under a future climate scenario.**
(PNG)

**S6 Fig. Classification of invaded vs. non-invaded strains out of 500,000 simulations from 100,000 parameterizations of the model.** Strains with <100 peak infections were considered not to have successfully invaded the population (left). Strains with <100 initial infections had an average of <1 infection over the course of the six-year simulation (right). Note the $\log_{10}$ scale of peak and mean infections.
(PNG)

**S7 Fig. Relationships between predicted and simulated outcomes in test data (200 independent HPAI simulations with randomly selected parameter values).** Model-predicted values show the expected values from generalized additive models, based on parameter values for each simulation. Error bars show 95% confidence intervals of the mean. Simulated values are actual outcomes from stochastic simulations. For continuous variables, the line shows a 1:1 relationship; for a perfect model, all points would fall on this line.
(PDF)

**S8 Fig. HPAI outcomes for strains with direct transmission only or environmental transmission only.** Points and error bars show means and 95% confidence intervals from raw data (error bars in (A) are too small to be visible). Density plots show distributions of raw data. All other parameters were randomly sampled from their distributions.
(PDF)

**S9 Fig. Effects of HPAI parameters on invasion probability.** Colors within each plot show the fitted probability of invasion, as estimated from a generalized additive model, as a function of the direct transmission rate (x) and another HPAI trait (y). All traits not shown are held at their median values, except $\beta_2$, which is held at $10^{-1.5}$. Plots show fitted values for a strain introduced at the breeding site on September 13.
(PDF)

**S10 Fig: Effect of introduction date and location on HPAI outcomes.** The y-axis shows the predictions from a generalized additive model, as a function of the combination of introduction date and location. Error bars shown 95% confidence intervals of the mean. All parameters not shown (i.e., HPAI traits) are held at their median values, except $\beta_2$, which is held at $10^{-1.5}$.
(PNG)

**S11 Fig. Sensitivity of persistence probability, infection prevalence, and mortality to mortality rates of HPAI-infected hosts ($\nu_2$) and the direct transmission rate ($\beta_2$).** (A) Prevalence and (B) persistence are low for strains that cause high mortality, especially when direct transmission rates are high. (C) The combination of direct transmission and high HPAI-induced mortality rates increase mortality, sometimes to the point of population extinction (mortality = 1). Each plot shows predicted outcomes from a generalized additive model that evaluated persistence duration as a function of viral traits and time/location of introduction, including pairwise interactions. All continuous

parameters not shown are held at their median values, except $\beta_2$, which is held at $10^{-1.5}$. Plots show fitted values for a strain introduced at the breeding site on September 13 (at the end of the breeding season).
(PDF)

**S12 Fig. Sensitivity of HPAI infection prevalence (A) and mortality (B) to heterospecific shedding and the environmental decay rate.** Note the low variation in values (i.e., range of color scale) in (A). Each plot shows predicted outcomes from a generalized additive model that modeled each outcome as a function of viral traits and time/location of introduction, including pairwise interactions. All continuous parameters not shown are held at their median values, except $\beta_2$, which is held at $10^{-1.5}$. Plots show fitted values for a strain introduced at the breeding site on September 13 (at the end of the breeding season).
(PDF)

**S13 Fig. The effect of climate change on invasion probability depends on the interaction between transmission mode, environmental decay rate, and the effects of climate.** Results are from a generalized linear model that modeled invasion as a function of strain and its interaction with climate impact (x-axis). Columns show direct transmission rates (values of $\beta_2$); only the three lowest values are shown because higher values of $\beta_2$ resulted in 100% invasion. Missing panels or points indicate strains without sufficient data to fit a model (e.g., very low invasion probability).Rows show shedding rates ($\omega_2$). Colors show environmental decay rates, which are inversely related to temperature sensitivity (i.e., strains with low decay rates are the most temperature sensitive).
(PDF)

**S14 Fig. The effect of climate change on peak infection prevalence depends on the interaction between transmission mode, environmental decay rate, and the effects of climate.** The y-axis shows the expected change in peak infection prevalence between 2020 and 2095. Results are from a generalized linear model that modeled invasion as a function of strain and its interaction with climate impact (x-axis). Columns show direct transmission rates (values of $\beta_2$); rows show shedding rates ($\omega_2$). Colors show environmental decay rates, which are inversely related to temperature sensitivity (i.e., strains with low decay rates are the most temperature sensitive). Missing panels or points indicate strains without sufficient data to fit a model (e.g., low invasion probability). Stars indicate the two strains shown in the main text.
(PDF)

**S15 Fig. The effect of climate change on outbreak size (i.e., peak infection prevalence at the breeding grounds) depends on the interaction between transmission mode, environmental decay rate, and the effects of climate.** The y-axis shows the expected change in outbreak size between 2020 and 2095. Results are from a linear model that modeled outbreak size as a function of strain and its interaction with climate impact (x-axis). Columns show direct transmission rates (values of $\beta_2$); rows show shedding rates ($\omega_2$). Colors show environmental decay rates, which are inversely related to temperature sensitivity (i.e., strains with low decay rates are the most temperature sensitive). Missing panels or points indicate strains without sufficient data to fit a model (e.g., low invasion probability). Stars indicate the two strains shown in the main text.
(PDF)

**S16 Fig. The effect of climate change on outbreak duration at the breeding grounds depends on the interaction between transmission mode, environmental decay rate, and the effects of climate.** The y-axis shows the expected change in outbreak duration between 2020 and 2095. Results are from a linear model that modeled outbreak duration as a function of strain and its interaction with climate impact (x-axis). Columns show direct transmission rates (values of $\beta_2$); rows show shedding rates ($\omega_2$). Colors show environmental decay rates, which are inversely related to temperature sensitivity (i.e., strains with low decay rates are the most temperature sensitive). Missing panels or points indicate strains without sufficient data to fit a model (e.g., low invasion probability). Stars indicate the two strains shown in the main text.
(PDF)

**S17 Fig. The effect of climate change on mean infection prevalence over six years depends on the interaction between transmission mode, environmental decay rate, and the effects of climate.** The y-axis shows the expected change in infection prevalence between 2020 and 2095. Results are from a linear model that modeled infection prevalence as a function of strain and its interaction with climate impact (x-axis). Columns show direct transmission rates (values of $\beta_2$); rows show shedding rates ($\omega_2$). Colors show environmental decay rates, which are inversely related to temperature sensitivity (i.e., strains with low decay rates are the most temperature sensitive). Missing panels or points indicate strains without sufficient data to fit a model (e.g., low invasion probability). Stars indicate the two strains shown in the main text.
(PDF)

**S18 Fig. The effect of climate change on mortality depends on the interaction between transmission mode, environmental decay rate, and the effects of climate.** The y-axis shows the expected change in mortality between 2020 and 2095. Results are from a linear model that mortality as a function of strain and its interaction with climate impact (x-axis). Columns show direct transmission rates (values of $\beta_2$); rows show shedding rates ($\omega_2$). Colors show environmental decay rates, which are inversely related to temperature sensitivity (i.e., strains with low decay rates are the most temperature sensitive). Missing panels or points indicate strains without sufficient data to fit a model (e.g., low invasion probability). Stars indicate the two strains shown in the main text.
(PDF)

**S1 Table. Parameters used in baseline simulations of low pathogenic and highly pathogenic avian influenza.** "Scale" gives the transformation used for parameter ranges and statistical modeling (e.g., $\beta_2$ was varied between $10^{-4}$ and 4.7 on a $\log_{10}$ scale). "Value used in HPAI models" defines the baseline LPAI strain. See Gonnerman *et al.* [50] for a full review of challenge studies.
(DOCX)

**S2 Table. Dates and locations of HPAI introduction in simulations measuring sensitivity of HPAI dynamics to viral traits. *t* corresponds to time (proportion of year) in model equations (see Supplementary Methods).**
(DOCX)

**S3 Table. Parameter values used in simulations of climate change.** All other parameters were held constant (see S1 Table for LPAI parameters). In all simulations, strains were introduced at the breeding grounds on May 27 ($t=0.4$). The viral decay rate provided ($\eta_2$) is the viral decay rate at 0°C and is inversely related to temperature sensitivity. Simulations used pariwise combinations of these three parameters, for a total of 120 strains; simulations were not run for strains where both $\beta_2 = 0$ and $\omega_2 = 0$. See S1 Table for parameter definitions. Other HPAI parameter values were fixed at: $\gamma_2 = 32.14$, $\nu_2 = 1.68$, $\psi_1 = 0.48$, $\psi_2 = 0.89$, $\zeta_2 = 2418$.
(DOCX)

**S4 Table. Model performance metrics for generalized additive models of HPAI outcomes.** Binary outcomes variables (invasion probability and persistence probability) were evaluated with area under the receiver operating curve (AUC) and continuous variables were evaluated with relative root mean squared error (rRMSE). Metrics are provided for both the training and test data.
(DOCX)

## Acknowledgments

Resources supporting this work were provided by the NASA High-End Computing (HEC) Program through the NASA Advanced Supercomputing (NAS) Division at Ames Research Center. Any use of trade, firm, or product names is for descriptive purposes only and does not imply endorsement by the U.S. Government.

## Author contributions

**Conceptualization:** Claire S. Teitelbaum, Diann J Prosser.

**Data curation:** Cory T Overton.

**Formal analysis:** Claire S. Teitelbaum.

**Investigation:** Claire S. Teitelbaum.

**Methodology:** Claire S. Teitelbaum, Michael L Casazza, Cory T Overton, Elliott L Matchett, Diann J Prosser.

**Project administration:** Claire S. Teitelbaum.

**Resources:** Diann J Prosser.

**Software:** Claire S. Teitelbaum.

**Supervision:** Michael L Casazza, Diann J Prosser.

**Visualization:** Claire S. Teitelbaum.

**Writing – original draft:** Claire S. Teitelbaum.

**Writing – review & editing:** Claire S. Teitelbaum, Michael L Casazza, Cory T Overton, Elliott L Matchett, Diann J Prosser.

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
