## [Decision Letter · Decision Letter 0]

22 Nov 2024

Dear Dr. Teitelbaum,

Please submit your revised manuscript within 60 days Jan 22 2025 11:59PM. If you will need more time than this to complete your revisions, please reply to this message or contact the journal office at ploscompbiol@plos.org. Samuel V. Scarpino

Academic Editor

PLOS Computational Biology

Tobias Bollenbach

Section Editor

PLOS Computational Biology

Feilim Mac Gabhann

Editor-in-Chief

PLOS Computational Biology

**Additional Editor Comments:**

**Journal Requirements:**

1) We have noticed that you have uploaded Supporting Information files, but you have not included a list of legends. Please add a full list of legends for your Supporting Information files after the references list.

2) Some material included in your submission may be copyrighted. According to PLOSu2019s copyright policy, authors who use figures or other material (e.g., graphics, clipart, maps) from another author or copyright holder must demonstrate or obtain permission to publish this material under the Creative Commons Attribution 4.0 International (CC BY 4.0) License used by PLOS journals. Please closely review the details of PLOSu2019s copyright requirements here: PLOS Licenses and Copyright. If you need to request permissions from a copyright holder, you may use PLOS's Copyright Content Permission form.

Potential Copyright Issues:

- Please confirm that you are the photographer of Figure1B, or provide written permission from the photographer to publish the photo under our CC BY 4.0 license.

- Figure 1A; Please provide a direct link to the base layer of the map (i.e., the country or region border shape) and ensure this is also included in the figure legend; and provide a link to the terms of use / license information for the base layer image or shapefile. We cannot publish proprietary or copyrighted maps (e.g. Google Maps, Mapquest) and the terms of use for your map base layer must be compatible with our CC BY 4.0 license.

3) We note that your Data Availability Statement is currently as follows: "All code written in support of this publication, as well as simulation input files and generated data, will be made publicly available at Zenodo prior to publication. Code and data are currently available for review at Zenodo: https://zenodo.org/records/13355279?preview=1&token=eyJhbGciOiJIUzUxMiJ9.eyJpZCI6IjBjZGM2YjZjLWFiZmYtNGUwNy04OThkLWIzYjk1OGE0YTE4MSIsImRhdGEiOnt9LCJyYW5kb20iOiJkY2Q2NDQ3ZTg1MjgwZGYzYmU1MzhmODVhN2VmNDc5ZCJ9.3Mr4pDx6p7qjQN_4ZceIXubnriGIEDylTUmmhZSXpmz5dEl6FHldhtEVHuhyjOZH4VMpgH3DX1-W6I34Jv1GGQ.". Please confirm at this time whether or not your submission contains all raw data required to replicate the results of your study. Authors must share the “minimal data set” for their submission. PLOS defines the minimal data set to consist of the data required to replicate all study findings reported in the article, as well as related metadata and methods (https://journals.plos.org/plosone/s/data-availability#loc-minimal-data-set-definition).

- The points extracted from images for analysis..

4) Please amend your detailed Financial Disclosure statement. This is published with the article. It must therefore be completed in full sentences and contain the exact wording you wish to be published. Please ensure that the funders and grant numbers match between the Financial Disclosure field and the Funding Information tab in your submission form. Note that the funders must be provided in the same order in both places as well.

**Reviewers' comments:**

Reviewer's Responses to Questions

**Comments to the Authors:**

Reviewer #1: Dear Authors,

Thank you for submitting your manuscript on the impacts of climate change on HPAI dynamics in migratory waterfowl. This is a critical area of research, and your study offers valuable insights into how viral transmission traits and environmental changes might interact to influence disease dynamics. However, after reviewing the study, I believe there are several areas where revisions and additional clarifications would strengthen the manuscript.

Abstract:

• The abstract starts with a broad statement about emerging infectious diseases, which is somewhat generic. Although it sets the stage, it would be stronger if it immediately focused on the topic at hand (avian influenza and climate change).

• The mechanistic model is briefly mentioned without enough context. It’s unclear what specific features of the model were novel or important to the study.

• While the focus is on transmission traits and climate change, the direct impact on migratory waterfowl populations is glossed over. The abstract should state more explicitly how these changes affect the overall ecology or population dynamics of waterfowl.

• The abstract ends on a general note about interactions between host and viral ecology. This could be more impactful by stating concrete findings or practical implications for wildlife management or conservation.

• Terms like "most important factors" and "slightly increases" are vague. The magnitude of the effects should be stated, even if approximate, to provide a clearer idea of the findings.

• The description of climate change impacts (e.g., warmer environments reducing viral survival) is not sufficiently detailed. The mechanisms by which climate change affects viral transmission should be clearer, especially considering the direct and indirect pathways mentioned.

• While the focus on transmission traits (e.g., contact and shedding rates) is important, the abstract does not address whether other potential factors (e.g., genetic adaptations or immunological responses) were considered.

Introduction:

• The introduction starts by discussing "emerging infectious diseases" in general, which dilutes the focus. While it's good to mention global relevance, the introduction could benefit from narrowing down to avian influenza and climate change impacts earlier to engage the reader with the study’s specific objectives.

• The introduction briefly mentions that climate change affects disease transmission via species distributions, but it lacks detailed exploration of the mechanisms by which climate change interacts with avian influenza transmission, persistence, or virulence.

• The literature cited (e.g., on climate change impacts or avian influenza dynamics) is presented without much critical analysis. For instance, how the cited models or studies shaped the current study or how gaps in those studies are being addressed is not discussed.

• The introduction cites general reviews (e.g., [4–6] on climate change effects) without discussing specific examples or how the present work builds upon or diverges from these previous studies.

• The decision to use a mechanistic model and the reasons for choosing specific parameters (e.g., migration phenology) are not well explained. There is no discussion of why greater white-fronted geese were chosen as the focal species and how representative they are of waterfowl or other species affected by HPAI.

• The introduction does not clearly state the hypotheses being tested or the expected outcomes. This leaves the reader guessing as to the study’s intended contributions.

Methods:

• The SIR model assumes homogeneous mixing within sites, which oversimplifies real-world conditions where migration and contact rates are likely heterogeneous. The validity of this assumption should be discussed, especially in relation to waterfowl’s complex social and migratory behaviors.

• The model distinguishes between direct and environmental transmission but doesn’t explain the criteria for classifying transmission modes in specific environmental contexts. This weakens the understanding of how robust the model is to various ecological settings.

• The model primarily focuses on traits like shedding and contact rates, which oversimplifies the full spectrum of viral dynamics. Other factors, such as viral mutation rates, antigenic drift, or immune system responses, are not considered, limiting the model’s ecological validity.

• The method relies heavily on viral decay rates being parameterized based on temperature, but this ignores the influence of other environmental factors such as humidity, UV radiation, or host population density, which can also significantly affect viral persistence.

• The choice of viral decay rate parameters from reanalysis products (MERRA-2) is presented without sufficient validation or comparison to empirical field data. The reliance on reanalysis data is a potential limitation, as such data may not accurately capture micro-environmental conditions where viral transmission occurs.

• The model’s assumption of unidirectional migration and fixed seasonal sites may not reflect the dynamic and variable migration behavior of geese in response to changing environmental conditions. This could result in inaccurate predictions of virus spread and persistence.

• Although the paper mentions changes in migration timing due to climate change, there is no consideration of potential mismatches between migration and food availability, which could drastically alter infection dynamics.

• While the methods mention stochastic processes, it is unclear how much variability was introduced into the model and whether sensitivity analyses were performed to assess how robust the results are to parameter uncertainty.

• The parameters used for migration and infection dynamics (e.g., decay rates, contact rates) are introduced with minimal justification or reference to empirical studies, making it difficult to assess whether the model is biologically realistic.

Results

• The study modeled 10,000 combinations of viral traits but without clear justification of how these traits realistically reflect natural viral diversity or evolutionary dynamics. For example, the models treat viral shedding rate (ω2) and transmission rate (β2) as independent, yet in reality, viral strains with higher shedding may also exhibit lower transmission due to immune response or other compensatory factors. This oversimplification may lead to unrealistic invasion outcomes.

• There is no discussion on the empirical basis for the range of viral traits (e.g., direct transmission rate or mortality rates). Without proper calibration against real-world data, the modeled viral strains may not represent the actual behavior of HPAI, making predictions less reliable.

• The results indicate that direct transmission rate (β2) was the most influential factor for invasion success. However, the model does not explore how environmental factors or host behavior might impact transmission in more complex scenarios (e.g., within-season migration patterns, mixed populations). This reductionist approach undermines the ecological complexities of migratory populations, where contact rates and social structure could play a significant role in disease spread.

• Cross-immunity (ψ1) is mentioned as having limited influence on invasion probability, but this contradicts other studies that suggest prior immunity can strongly influence viral transmission dynamics. The limited exploration of cross-immunity seems like a missed opportunity, especially since cross-immunity may be a key factor in HPAI evolution. Additionally, the study focuses solely on the migratory goose population, yet HPAI is known to infect multiple hosts, including non-migratory species. The absence of a multi-host model could result in underestimating the risk of persistent HPAI transmission and endemicity, particularly in mixed species environments like wetlands.

• The study defines an invasion as ≥100 infections, but this threshold is arbitrary and may not correspond to a meaningful epidemic. In smaller populations, 100 infections could represent a significant fraction, while in larger populations, it may be negligible. This rigid threshold could skew the interpretation of results, especially when considering migratory populations where flock sizes fluctuate seasonally.

• While the model considers environmental persistence (η2), the results show weak effects of environmental persistence on viral outcomes. This could be due to oversimplified assumptions regarding how temperature influences viral viability in the environment. The model does not account for factors like water salinity or UV exposure, which are known to affect virus survival, especially in aquatic habitats. This could lead to an underestimation of the role of environmental reservoirs in HPAI dynamics.

• The study's approach to temperature sensitivity lacks granularity; it uses broad categories (e.g., temperature-sensitive strains) without detailing how small temperature changes influence viral persistence. The nonlinear effects reported should be more rigorously explained.

• The results mention that persistence duration was insensitive to direct transmission rates but was influenced by mortality rates (ν2). This finding contradicts well-established principles in disease ecology, where higher transmission often leads to longer persistence due to sustained chains of infection. The lack of discussion on why direct transmission did not significantly impact persistence is a major weakness, as it leaves a key question about disease longevity unresolved.

• The study models its outcomes using generalized additive models (GAMs), but it does not provide enough detail on model diagnostics, such as goodness of fit or validation against independent data sets. Without knowing how well the model predicts observed outcomes, it is difficult to assess the accuracy of the predictions. Additionally, GAMs can obscure underlying causal relationships by overfitting the data, especially when nonlinear effects are involved.

Discussion

• The discussion suggests that climate change will have opposing effects on HPAI dynamics depending on viral persistence or migration timing. This binary framing oversimplifies the complex interaction between climate, bird behavior, and virus ecology. For instance, it fails to account for how climate change might impact not just the timing of migration but also the geographic range of migratory species, potentially introducing HPAI to new hosts or ecosystems.

• Additionally, the discussion assumes that warmer climates will uniformly reduce viral persistence, but this may not hold in regions where other factors, like humidity, rainfall, or habitat degradation, might maintain environmental viral reservoirs despite increased temperatures.

• The discussion briefly acknowledges that multi-host dynamics play a role in viral persistence (lines 348–350), but this point is underdeveloped. Given that influenza viruses often spill over between species, the model’s failure to incorporate multi-host interactions limits its applicability. This omission is particularly relevant for understanding how species with different migration patterns or immune responses (e.g., ducks, shorebirds) could influence the overall persistence and spread of HPAI.

• The authors discuss how shifts from direct to environmental transmission could promote longer outbreaks or HPAI endemicity, but they do not explore the evolutionary implications of this shift. For example, will future strains become more temperature-resistant, or could selection pressure favor strains with higher transmission rates? A more detailed evolutionary perspective would strengthen the argument about how HPAI dynamics might change in response to environmental or host changes.

• The discussion does not adequately address the study’s limitations, such as the use of simplified host demographics (ignoring age structure and interspecies interactions), the arbitrary invasion threshold, or the potential bias in model parameter ranges. Without explicitly acknowledging these weaknesses, the predictive power of the model is overstated, reducing confidence in its utility for forecasting real-world HPAI dynamics.

• While the authors call for more experimental studies on AIVs (lines 367-372), they do not sufficiently integrate existing empirical data into their model. For example, temperature sensitivity and shedding rates are mentioned but not well-supported by field or lab-based studies. The model’s predictions would be more credible if it were more closely tied to real-world observations of HPAI outbreaks.

Reviewer #2: The study by Teitelbaum and co-authors presents a timely and well-executed study that advances understanding of avian influenza transmission under climate change scenarios using Greater White-fronted Geese as a model organism. Using a mechanistic model that integrates migration phenology and HPAI infection dynamics, the authors explore critical aspects of transmission that represent large knowledge gaps for the influenza field. The integration of direct and indirect effects of climate change into the compartmental model is particularly novel and insightful. The study lends support to the hypothesis that climate change will have opposing effects on HPAI transmission in migratory waterfowl due to reduced viral persistence in a warming climate, but also increased mean prevalence and mortality related to advancing spring migration. This Overall, I thought the study was thoughtful in paying attention to biologically-relevant variables and parameters that drive influenza infection at ecological scales, and am excited to see the impact that this work will have on the influenza and disease ecology field at-large.

I have a few recommendations to help strengthen the study, mostly with the goal of embedding the viral kinetics in empirical data derived from field or experimental studies:

- The consideration of two influenza phenotypes: LPAI and HPAI, and their cross-reactivity is insightful and demonstrates the authors have current working knowledge of influenza biology and the relevance of antigenically different strains. The endemic circulation of LPAI has been postulated as a major factor explaining the heterogeneity in disease outcome within and between avian hosts during the current HPAI outbreak (although never explicitly tested). However, cross-reactivity is treated as a binary variable: low and high, which is an over-simplification that may undermine the relevance of the model. For example, gulls in which H13 and H16 circulates endemically would offer little or no immunological protection against HPAI H5. I wondered if this could be characterized as a continuous trait measured by antigenic distance of the HA gene segment. Pairwise interactions between the 16 LPAI subtypes vs 1 HPAI subtype could be quantified and used to derive an underlying distribution of cross-protective immunity that would add resolution to the model, and make the results more biologically meaningful.

- Climate change is modeled as the mean annual temperature across each of the 4 sites of the annual cycle. This approach treats climate change as a gradual temperature change, without considering the increased variability in temperature at each of the 4 sites. Could the authors comment on how accounting for the two temporal scales of climate change - rapid (extreme temperature fluctuations) and gradual (mean temperature increases) might change the outcome of the model. For instance, the process of freeze-thawing can lead to degradation of the viral particle, which may further reduce viral persistence especially at habitats that exist on or near the 0 degree threshold.

Minor comments:

- The use of the term ‘pathogen invasion’ isn’t commonly used in influenza literature, or at least not to my knowledge. I can see it relates to the Rohani paper from 2006. I’m guessing this is functionally the same as introduction of the virus into a new population. Might be worth clarifying that in the introduction.

Reviewer #3: This paper provides a comprehensive assessment of the role that high pathology avian influenza (HPAI) could play in the population dynamics of a migratory bird. Using a simulation approach, the different intrinsic traits of the HPAI strain, changes in climate, and migratory behaviors of the species can be assessed in concert to assess these factors for overall importance and plan for likely future scenarios. Overall, the modeling approach is robust, well-documented, and well-executed. This was a significant effort that was undertaken with a consistent logic and purpose.

There are some areas this paper can improve upon. This paper has a fair bit going on in it with all the different scenarios tested and communicating the results of this paper is challenging. With so many scenarios tested and simulations run, there are some difficulties in communicating complex and interacting results. I would prefer to see more figures in the main paper to help the reader understand the patterns. I spent a fair bit of time in the SI when reviewing the paper and I found the figures there were very helpful to understanding the results. I would bring in a few of the SI figures into the main paper. S7 provides overall summary statistics for the effort that would be helpful for the readers to see, and S9 provides support for a key component of the paper. Maybe even Figure 4 is useful for communicating the overall rates of strain invasions. But more information and data visualization are needed in the main paper. Further, the main paper figures also needed some updates. I provide more details in the line-by-line comments there.

The second larger point is what you should do for all parameters and scenarios that were tested. This paper has some issues with information overload, and a few changes could help the reader out. First, I think a summary table or figure describing the overall importance of each parameter would be helpful. I realize this would be challenging given all the complexities here, but something to provide an overall assessment of significant relationships or interactions across the parameters would go a long way. Furthermore, there are knowledge gaps in the parameters chosen for this paper. Some parameters are well-known, and others are not. More detail on when a parameter value incorporated into the simulation study is precautionary or well-supported by the literature would help direct the readers' attention. Further, you could combine your assessments of current knowledge with model sensitivity across all the parameters to determine which values we need more information on. These types of studies are only as good as the data that go into them, and directing future research to areas where more accuracy is required would be a valuable contribution.

Even with a few issues, I found this paper very well done and interesting to read. Thanks for the contribution.

Line-by-line comments:

26-27. This sentence is a bit awkward. What is ‘those’ specifically? Also, pointing out that more deadly outbreaks can cause significant mortality doesn’t seem helpful or specific.

197. SSP 3-7.0 is on the higher end of climate predictions, I don’t think it can be solely described as moderate.

217. I would change this to ‘...all could be defined on the interval of [0, 1];...’

247. Are transmission rates from 0.1 to 1 considered low to intermediate? It should be something like ‘low to maximum’ at least. This doesn’t seem like a useful category of values to me.

309. I think the different symbols and transparencies are difficult to read in Figure 3. The y-axis labeling is confusing, and it's difficult to parse what matters. I would try changing the symbology or a different type of figure altogether.

354. This finding should be more clearly shown in the main paper.

369. This sentence seems like it should be an entire paragraph in the discussion. Given that you are using fairly wide ranges for many of the variables because of uncertainty in model parameters, can the parameter inputs be evaluated in terms of prior knowledge? Table S1 covers how each of the inputs was derived, but it would be useful to know which parameters had limited knowledge and which were more strongly informed. Further, this could be carried over to something like Figure 2, where the more likely parameter values could be identified to direct our attention.

389. You don’t need ‘historically’ in this sentence.

**Have the authors made all data and (if applicable) computational code underlying the findings in their manuscript fully available?**

Reviewer #1: None

Reviewer #2: Yes

Reviewer #3: Yes

PLOS authors have the option to publish the peer review history of their article (what does this mean? ). If published, this will include your full peer review and any attached files.

**Do you want your identity to be public for this peer review?** For information about this choice, including consent withdrawal, please see our Privacy Policy .

Reviewer #1: **Yes: ** Mohammed Rohaim

Reviewer #2: No

Reviewer #3: No

**Figure resubmission:**
---

## [Decision Letter · Decision Letter 1]

13 Jul 2025

PCOMPBIOL-D-24-01415R1

Host responses and viral traits interact to shape the impacts of climate warming on highly pathogenic avian influenza in migratory waterfowl

PLOS Computational Biology

Dear Dr. Teitelbaum,

Thank you for submitting your manuscript to PLOS Computational Biology. After careful consideration, we feel that it has merit but does not fully meet PLOS Computational Biology's publication criteria as it currently stands. Therefore, we invite you to submit a revised version of the manuscript that addresses the points raised during the review process.

Please submit your revised manuscript within 30 days Sep 12 2025 11:59PM. If you will need more time than this to complete your revisions, please reply to this message or contact the journal office at ploscompbiol@plos.org. Please include the following items when submitting your revised manuscript:

We look forward to receiving your revised manuscript.

Kind regards,

Samuel V. Scarpino

Academic Editor

PLOS Computational Biology

Tobias Bollenbach

Section Editor

PLOS Computational Biology

**Additional Editor Comments:**

I believe that the remaining comments/questions from R1 can be addressed with additions to the text of the manuscript without more simulations, etc.. Please pay careful attention to ensure all of R1's comments are addressed in the revision.

**Journal Requirements:**

**Reviewers' comments:**

Reviewer's Responses to Questions

**Comments to the Authors:**

Reviewer #1: This study investigates how climate change impacts the dynamics of highly pathogenic avian influenza (HPAI) in migratory waterfowl, focusing on the interplay between viral traits (e.g., transmission modes, environmental persistence) and host ecology (e.g., migration phenology). Using a mechanistic model, the authors simulate HPAI outbreaks under current and future climate scenarios, revealing that climate warming can either amplify or mitigate HPAI impacts depending on whether it affects viral decay rates or bird migration timing.

Major Concerns:

• The model assumes homogeneous mixing within sites, ignoring social structures (e.g., familial groups in geese). How might this bias transmission dynamics?

• The study focuses on Pacific greater white-fronted geese. Would results generalize to other waterfowl, especially species with different migration patterns or tolerance to HPAI?

• Only two SSP scenarios (2-4.5 and 3-7.0) are tested. Could more extreme or intermediate scenarios yield different insights?

• The ranges for key parameters (e.g., direct transmission rate β2β2) are broad due to limited empirical data. How sensitive are conclusions to these uncertainties?

• Cross-immunity (ψ1,ψ2ψ1,ψ2) is fixed, but real-world variability in immunity across strains/hosts could alter dynamics. Was this explored?

• Lines 527–529 suggest environmental transmission leads to longer outbreaks, but results (e.g., Figure 3) show direct transmission drives higher peak prevalence. Clarify this discrepancy.

• The abstract states climate warming "increases HPAI impacts" (line 24), but results show opposing effects (e.g., reduced environmental persistence). Reconcile these statements.

• How were migration timings validated against empirical data? Were interannual variations in migration considered?

• The inverse relationship between decay rate and temperature sensitivity (line 214) is critical but relies on a single study (Handel et al.). Are there conflicting data?

• The 9-day advancement in spring migration (line 469) seems small relative to projected temperature increases. Is this biologically plausible?

• Microhabitat refugia (e.g., shaded wetlands) could buffer temperature effects. Were these accounted for?

• Figure 1: Missing panel labels (A–F) in the text layer. Ensure consistency with the caption.

• Figure 3: Y-axis labels for "Proportion of simulations" are cut off in panels A and B.

• Table S1: Clarify units for parameters like β2β2 (is it per day?). Include citations for parameter ranges.

• Table S3: The note says β2=0β2=0 and ω2=0ω2=0 simulations were excluded, but why? This limits understanding of purely immunity-driven dynamics.

• Typos:

o Line 38: "prior infection provides little immunityviruses" → missing space.

o Line 213: "imbeddinged": "embedded".

o Line 527: "replicated LPAI infection dynamics in geese [57], even while ignoring excluding": redundant phrasing.

o Line 24–25: "climate warming has opposing impacts... via viral persistence and bird migration phenology": rephrase for clarity (e.g., "via opposing effects on viral persistence and migration phenology").

o Line 348: "We modeled 10,000 potential combinations": "We simulated".

Reviewer #2: I believe the paper is fit for publication and the authors have addressed all my concerns

Reviewer #3: I thought your changes were appropriate for my comments as well as the other reviewers.

**Have the authors made all data and (if applicable) computational code underlying the findings in their manuscript fully available?**

Reviewer #1: None

Reviewer #2: Yes

Reviewer #3: Yes

PLOS authors have the option to publish the peer review history of their article (what does this mean? ). If published, this will include your full peer review and any attached files.

**Do you want your identity to be public for this peer review?** For information about this choice, including consent withdrawal, please see our Privacy Policy .

Reviewer #1: No

Reviewer #2: No

Reviewer #3: **Yes: ** Evan Adams

**Figure resubmission:**
---

## [Editor Report · Decision Letter 2]

19 Aug 2025

Dear Ms. Teitelbaum,

We are pleased to inform you that your manuscript 'Host responses and viral traits interact to shape the impacts of climate warming on highly pathogenic avian influenza in migratory waterfowl' has been provisionally accepted for publication in PLOS Computational Biology.

Best regards,

Samuel V. Scarpino

Academic Editor

PLOS Computational Biology

Tobias Bollenbach

Section Editor

PLOS Computational Biology

---

## [Editor Report · Acceptance letter]

PCOMPBIOL-D-24-01415R2

Host responses and viral traits interact to shape the impacts of climate warming on highly pathogenic avian influenza in migratory waterfowl

Dear Dr Teitelbaum,

I am pleased to inform you that your manuscript has been formally accepted for publication in PLOS Computational Biology. Your manuscript is now with our production department and you will be notified of the publication date in due course.

With kind regards,

Zsofia Freund
